# Variation in Population and Solvents as Factors Determining the Chemical Composition and Antioxidant Potential of *Arctostaphylos uva-ursi* (L.) Spreng. Leaf Extracts

**DOI:** 10.3390/molecules27072247

**Published:** 2022-03-30

**Authors:** Piotr Sugier, Łukasz Sęczyk, Danuta Sugier

**Affiliations:** 1Department of Botany, Mycology and Ecology, Institute of Biological Sciences, Maria Curie-Skłodowska University, 19 Akademicka Street, 20-033 Lublin, Poland; 2Department of Industrial and Medicinal Plants, University of Life Sciences in Lublin, 15 Akademicka Street, 20-950 Lublin, Poland; lukasz.seczyk@up.lublin.pl (Ł.S.); danuta.sugier@up.lublin.pl (D.S.)

**Keywords:** bearberry, *Uvae ursi folium*, arbutin, hydroquinone, natural antioxidants

## Abstract

The bearberry *Arctostaphylos uva-ursi* (L.) Spreng. has a long history of ethnopharmacological use. This species has been used in folk medicine for centuries as a rich source of raw material abundant in secondary metabolites and is important for medicinal and pharmacological purposes. The plant is a source of herbal material—*Uvae ursi folium*, which is highly valued and sought by pharmaceutical and cosmetic industries. The studied bearberry leaves can be classified as a suitable herbal material for use in pharmacy; therefore, the investigated populations can be a potentially valuable source of plant material for cultivation and can be used in in vitro cultures and in biotechnological processes. The objective of this study was to characterize the variability of the phytochemical composition and antioxidant activity of water and ethanol bearberry extracts from raw material collected from different natural populations. In each of the twelve *A. uva-ursi* sites, three leaf samples were collected and analyzed. The water extracts from bearberry leaves were characterized by similar concentration of arbutin (77.64–105.56 mg g^−1^) and a significantly higher concentration of hydroquinone (6.96–13.08 mg g^−1^) and corilagin (0.83–2.12 mg g^−1^) in comparison with the ethanol extracts −77.21–103.38 mg g^−1^, 10.55–16.72 mg g^−1^, 0.20–1.54 mg g^−1^, respectively. The concentration of other metabolites in the water extracts was significantly lower in comparison with the ethanol extracts. In the case of the water extracts, a significant effect of not only total phenolic compounds, but also hydroquinone on the antioxidant parameters, was observed, which indicates the solvent-related activity of these metabolites. Therefore, it is suggested that special attention should be paid to the concentration of not only arbutin, but also hydroquinone in *Uvae ursi folium*. The latter metabolite serving a very important function as an active bearberry ingredient should be controlled not only in alcoholic extracts but also in water extracts, since bearberry leaves are applied as infusions and decoctions. The results presented in this paper can contribute to appropriate selection of plant material for pharmaceutical, cosmetic, and food industries, with special emphasis on the antioxidant activity of different types of extracts.

## 1. Introduction

The use of medicinal plants as a source of very valuable therapeutic substances has been growing rapidly in the world due to the increasing demand for natural secondary metabolites [1,2,3]. In the last decade, a trend towards replacing the use of synthetic antioxidants with natural compounds with antioxidative activity in food industry applications is observed [4,5]. Effective nontoxic natural compounds are searched for in the natural environment. New sources of secondary metabolites of medicinal plants are being widely sought [6,7], and introduction of medicinal plants into field conditions is carried out to obtain chemically interesting standardized raw material in controlled conditions [6,7,8]. However, there are rare attempts to search for chemically valuable bearberry ecotypes in natural habitats and to analyze variations within and among natural bearberry populations [9,10].

*Arctostaphylos uva-ursi* (L.) Spreng. is a species with a high concentration of phenolic compounds, especially arbutin, which is the primary bioactive compound in this plant. Arbutin was also detected in other plants, e.g., *Bergenia* spp. [11], *Pyrus* spp. [12], *Vaccinium* spp. [4], *Arbutus unedo* [13], and *Origanum* spp. [14]; nevertheless, the bearberry is regarded as the main natural source of arbutin to be used for phytoterapy purposes. *A. uva-ursi* is an endangered and protected species in many European countries [15,16,17,18,19]. Therefore, the identification of the chemical potential especially of such an endangered species as bearberry is necessary. This will facilitate the use of the most interesting ecotypes from the point of view of the composition of secondary metabolites in field cultivation, in in vitro cultures, and in biotechnological processes. Simultaneously, such investigations should help to reduce the pressure exerted by harvesting in unprotected areas, where a decrease in the regenerative capacity of *A. uva-ursi* populations is observed [20,21].

The bearberry leaves (BL) have been used in folk medicine for centuries as an interesting source of secondary metabolites and is important for medicinal and pharmacological purposes. *Uvae ursi folium* is very often sought by pharmaceutical and cosmetic industries [22,23]. Its health benefits are provided by the compounds comprised in its valuable composition. Recently, the composition of the secondary metabolites in BL has been intensively investigated [24,25,26]. In addition to the aforementioned arbutin, the chemical profile of BL is characterized by the presence of gallic acid, ursolic acid, tannic acid, p-coumaric acid, galloylarbutin, gallotannins, quercetin, kaempferol, penta-O-galloyl-β-d-glucose, corilagin, picein, hyperoside, and many other compounds [4,24]. *Uvae ursi folium* extracts are remedies for several diseases, e.g., diuresis [25,26], and have been used as skin-whitening factors and antioxidant agents in food packaging [27,28,29]. The main component arbutin is a skin depigmenting agent with antimelanogenic and antioxidant properties [30]. *Arctostaphylos uva-ursi* leaf extracts (ALE) are characterized by antioxidant, antimicrobial, and antiproliferative activity [27,31,32,33,34].

*Arctostaphylos uva-ursi* is a well-known traditional herbal plant used in the treatment of urinary tract infections. The antiseptic and diuretic activity of this metabolite can be attributed to hydroquinone, which is obtained by hydrolysis of arbutin [35]. Naturally occurring hydroquinone was found in certain plants following its release from arbutin upon plant β-glucosidase activity [36]. Although free hydroquinone occurs naturally in the leaves of various medicinal plant species [37], the presence of this metabolite in BL has been demonstrated extremely rarely. In herbal preparations, it is recognized as an active substance at the site of action (urinary tract) and is crucial for therapeutic activity. Hydroquinone has hepatotoxic, nephrotoxic, and genotoxic potential. This metabolite has been found in plants not only as arbutin but also in the free form [38]. Moreover, it has been suggested that its application in the treatment of human urinary infections must be controlled through the intake of both arbutin and hydroquinone in the diet [37].

Phenolic compounds in BL have been detected by different modern techniques, quantified, and characterized, and the antioxidant activity of these compounds has been evaluated [24,27,31,32,39]. Additionally, each study used different extraction methods and/or different extraction solvents [9,24,26,27,29,31,32,40]. To the best of our knowledge, the use of water as a solvent has not been investigated to date. Hence, there is sparse knowledge of the phytochemical characteristics of extracts and functions of hydrophilic secondary metabolites, especially hydroquinone. Therefore, the use of different solvents, including water, allows a more complete description of the raw material extracts. This is especially important in the context of using infusions and decoctions of *Uvae ursi folium* and the biological activity of hydroquinone. Therefore, the objective of this study was to characterize the variability of the phytochemical composition and antioxidant activity of bearberry in water and ethanol extracts of raw material collected from different natural populations. Additionally, revealing also the role of hydrophilic substances in bearberry leaf extracts was determined. The results presented in this paper can contribute to appropriate selection of plant material for pharmaceutical, cosmetic, and food industries, with special emphasis on the antioxidant activity of different types of extracts.

## 2. Results and Discussion

### 2.1. Characteristics of Secondary Metabolites

The two-way ANOVA results showed a statistically significant impact of the population (F = 44.6, *p* < 0.001), extraction method (F = 7905.6, *p* < 0.001), and their interaction (F = 9.8, *p* < 0.001) on the total phenolic concentration. The water extracts (WE) from the BL exhibited a wide variation in the total phenolic concentration, which ranged from 165.63 mg GAE g^−1^ to 214.84 mg GAE g^−1^. Similarly, the total phenolic concentration in the ethanol extracts (EE) ranged from 258.03 mg GAE g^−1^ to 298.52 mg GAE g^−1^ (Figure 1). The statistical analysis showed differences in the mean total phenolic concentration values between the populations and between the extracts. Clearly visible was the difference in the total phenolic concentration between the two types of extracts. The total phenolic concentration in the EE was ca. 30% higher than in the WE and two-fold higher than in the EE of BL studied in Spain [9].

The results of the statistical analyses showed a significant impact of the population (F = 30.5, *p* < 0.001) and extraction method (F = 952.2, *p* < 0.001) on the total flavonoid concentration. The interaction of the main factors was not confirmed (F = 1.2, *p* = 0.346). The analysis of the WE from the BL showed a wide variation in the total flavonoid concentration ranging from 2.36 mg QE g^−1^ to 3.09 mg QE g^−1^ and in the EE, i.e., from 3.21 mg QE g^−1^ to 3.88 mg QE g^−1^ (Figure 2). The statistical analysis showed statistically significant differences in the mean total flavonoid concentration values between the populations and between the extracts. Clearly visible was the difference in the total flavonoid concentration between the two types of extracts. The total flavonoid concentration in the EE was over 20% higher than in the WE.

The two-way ANOVA results showed a statistically significant impact of the population (F = 633.9, *p* < 0.001) on the concentration of arbutin. The effect of the extraction method (F = 1.9, *p* = 0.173) and interaction of the main factors (F = 1.6, *p* = 0.228) was not confirmed statistically. The analysis of the extracts from the BL showed a variation in the arbutin concentration, i.e., from 77.64 mg g^−1^ to 105.56 mg g^−1^ in the WE and from 77.21 mg g^−1^ to 103.38 mg g^−1^ in the samples extracted by ethanol (Figure 3). Although the concentration of phytoconstituents depends on the extraction method [12,41,42,43], no arbutin variation was observed in the present study. Both studied extracts are polar; therefore, the small difference in the polarity did not influence the arbutin concentration.

The arbutin concentration was similar in the WE and EE in 11 populations (Figure 3). There were differences in the mean values of this substance between the samples collected in the different populations. The arbutin concentration in BL must be at least 70 mg g^−1^ to recognize the plant as herbal material [44]. The present results indicate that the bearberry material from all the studied populations meets the European Pharmacopeia requirements regarding the concentration of this metabolite. Arbutin is a characteristic metabolite for many Ericaceae plant species: *Vaccinium myrtillus*, *V. uliginosum*, and *V. vitis-idaea*; however, its highest concentration was detected in *A. uva-ursi* [4]. The concentration of arbutin in the present studies is similar to the concentration range reported from natural bearberry habitats in the Iberian Peninsula [9,45,46].

The results of the statistical analyses showed a significant impact of the population (F = 274.6, *p* < 0.001), extraction method (F = 4108.2, *p* < 0.001), and their interaction (F = 11.2, *p* < 0.001) on the concentration of hydroquinone in the BL. The analysis of the WE showed a wide variation in hydroquinone ranging from 10.55 mg g^−1^ to 16.72 mg g^−1^ and in the EE, i.e., from 6.96 mg g^−1^ to 13.08 mg g^−1^ (Figure 4). Clearly visible was the difference in the concentration of hydroquinone between the two types of extracts. The hydroquinone concentration in the WE was over 30% higher than in the EE. Our study has shown that the concentration of hydroquinone can be high in raw material after drying, even before the manufacturing process of raw materials. Although arbutin is the major pharmacological active constituent of the analyzed plant material, experimental studies have revealed that the whole extract is responsible for the global pharmacological action [38]. Therefore, during analyses of bearberry raw material, attention should be paid to the hydroquinone concentration in the WE and EE. Moreover, it should be remembered that exposure to microorganisms or ultraviolet radiation during storage and use of cosmetic products has the potential to generate hydroquinone [47,48]. At present, it is not known whether storage and exposure of bearberry raw material to ultraviolet radiation change its chemical composition.

The two-way ANOVA results showed a statistically significant impact of the population (F = 258.5, *p* < 0.01), extraction method (F = 2842.2, *p* < 0.01), and their interaction (F = 136.2, *p* < 0.01) on the concentration of methylarbutin in the BL. The analysis of the extracts showed a variation in the methylarbutin concentration, i.e., from 0.45 mg g^−1^ to 3.97 mg g^−1^ in the WE and from 0.94 mg g^−1^ to 9.76 mg g^−1^ in the samples extracted by ethanol (Figure 5). In some populations (1, 3, 5; Figure 5), the concentration of this metabolite was even several times greater in the EE than in the WE.

The results of the statistical analyses showed a significant impact of the population (F = 223.1, *p* < 0.01), extraction method (F = 803.4, *p* < 0.01), and their interaction (F = 39.3, *p* < 0.01) on the concentration of penta-O-galloyl-β-d-glucose (PGG) in the BL. The analysis of the WE showed a wide variation in PGG ranging from 1.90 mg g^−1^ to 7.48 mg g^−1^ and in the EE, i.e., from 3.82 mg g^−1^ to 11.72 mg g^−1^ (Figure 6). Penta-O-galloyl-β-d-glucose, i.e., gallotannin, is a polyphenolic compound occurring naturally in several medicinal plants: *Acer truncatum* [49], *Fomitella fraxinea* [50], *Paeonia suffruticosa* [51], *Schinus terebinthifolius* [52], and *Rhus* spp. [53,54]. It exhibits multiple biological activities with considerable potential to be used in the therapy and prevention of several major diseases, including cancer and diabetes [49,55,56]. 

The two-way ANOVA results showed a statistically significant impact of the population (F = 282.5, *p* < 0.001), extraction method (F = 125.5, *p* < 0.001), and their interaction (F = 73.8, *p* < 0.001) on the concentration of picein in the BL. The analysis of the ALE showed a variation in the picein concentration, i.e., from 1.13 mg g^−1^ to 1.98 mg g^−1^ in the WE and from 0.69 mg g^−1^ to 1.75 mg g^−1^ in the samples extracted by ethanol (Figure 7). This picein concentration in the EE is lower than in raw material taken from heathland populations, where it exceeded 2.5 mg g^−1^ [10]. Picein is a natural antioxidant and can serve a function of a potent neuroprotectant [57]. 

The results of the statistical analyses showed a significant impact of the population (F = 452.2, *p* < 0.01), extraction method (F = 13354.7, *p* < 0.01), and their interaction (F = 225.3, *p* < 0.01) on the concentration of corilagin in the BL. The analysis of the WE showed a wide variation in corilagin ranging from 0.83 mg g^−1^ to 2.12 mg g^−1^; in the EE, it ranged from 0.20 mg g^−1^ to 1.54 mg g^−1^ (Figure 8). There was a clearly visible difference in the concentration of corilagin between the two types of extracts. In the WE, it was even six time higher than in the EE. Corilagin is characterized by a broad spectrum of biological and therapeutic properties, e.g., anti-inflammatory [58], antioxidant [59], and hepatoprotective [60] activities. 

The two-way ANOVA results showed a statistically significant impact of the population (F = 212.5, *p* < 0.001), extraction method (F = 650.5, *p* < 0.001), and their interaction (F = 7.4, *p* < 0.001) on the concentration of hyperoside in the BL. The analysis showed a variation in the picein concentration, i.e., from 4.23 mg g^−1^ to 6.19 mg g^−1^ in the WE and from 4.95 mg g^−1^ to 7.48 mg g^−1^ in the EE (Figure 9). The difference in the concentration of hyperoside between the two types of extracts was evident. The concentration of this metabolite in the EE was over 15% higher than in the WE. The hyperoside concentrations presented in this paper are significantly higher than those reported for heathland populations [10]. Hyperoside is an active compound found in plants of the genera *Hypericum* and *Crataegus* exhibiting antioxidant, anticancer, and anti-inflammatory activities [61,62,63].

The results of the PCA ordination of the water and ethanol ALE are presented in Figure 10. The eigenvalues of axis 1 (4.08) and axis 2 (1.58) show the presence of two main gradients (Table 1). The first two axes explain 62.88% of the variability (45.38%–axis 1, 17.50%–axis 2). The total phenolic concentration, total flavonoid concentration, and methylarbutin in the analyzed extracts are clearly positively correlated with axis 1, whereas corilagin and hydroquinone are correlated negatively. The concentration of picein and abutin are positively correlated with axis 2. Axis 1 shows an increase in the concentration of total phenolic concentration, total flavonoid concentration, and methylarbutin and a decrease in the concentration of corilagin and hydroquinone are observed, which implies that primarily the concentration of these metabolites determines the differentiation of the chemical composition of the WE and EE. In Figure 10, there are two clearly separated groups: a group of leaf samples extracted by water on the left and a group of samples extracted by ethanol on the right. The first group is characterized by the highest concentration of hydroquinone and corilagin and the lowest concentrations of total phenolic compounds, total flavonoids, methylarbutin, and PGG in relation to the EE.

The comparison of the values of the particular characteristics generally shows differences between the populations and between the WE and EE (Figure 1, Figure 2, Figure 3, Figure 4, Figure 5, Figure 6, Figure 7, Figure 8, Figure 9 and Figure 10, Table 1). The EE of the ALE are characterized by significantly higher concentrations of TPC, TFC, mARB, PGG, PIC, and HYP and significantly lower concentrations of HQ and COR in comparison with the BL extracted by water. The plant species composition, light conditions, altitude, precipitations, and radiation can change the chemistry of BL [9]. The present results show that not only the habitat and population characteristics but also the use of different solvent contribute to better characterization of bearberry raw material.

### 2.2. Antioxidant Activity

The two-way ANOVA results showed a statistically significant impact of the population (F = 79.4, *p* < 0.001), extraction method (F = 19203.2, *p* < 0.001), and their interaction (F = 5.1, *p* < 0.001) on the ABTS^•+^ scavenging activity of the analyzed ALE. The range of ABTS scavenging was from 173.48 mg TE g^−1^ to 319.22 mg TE g^−1^ and from 503.52 mg TE g^−1^ to 643.71 mg TE g^−1^ in the WE and EE, respectively (Table 2). The results of the statistical analyses showed a significant impact of the population (F = 39.9, *p* < 0.01), extraction method (F = 2731.1, *p* < 0.01), and their interaction (F = 42.6, *p* < 0.01) on the DPPH^•^ scavenging activity of the analyzed ALE. The variations in DPPH scavenging ranged from 490.59 mg TE g^−1^ to 821.78 mg TE g^−1^ and from 377.35 mg TE g^−1^ to 516.28 mg TE g^−1^ in samples extracted by water and ethanol, respectively (Table 2). The statistical analyses showed a significant impact of the population (F = 39.6, *p* < 0.001), extraction method (F = 7718.2, *p* < 0.001), and their interaction (F = 3.8, *p* < 0.001) on the reducing power of the analyzed ALE. The range of the reducing power parameter was from 245.37 mg TE g^−1^ to 296.20 mg TE g^−1^ and from 353.26 mg TE g^−1^ to 404.89 mg TE g^−1^ in the WE and EE, respectively (Table 2). The two-way ANOVA results showed a statistically significant impact of the population (F = 48.9, *p* < 0.01), extraction method (F = 274.3, *p* < 0.01), and their interaction (F = 48.8, *p* < 0.01) on the chelating ability of the analyzed extracts. The variations in chelating ability ranged from 3.71 mg EDTA g^−1^ to 4.96 mg EDTA g^−1^ and from 3.56 mg EDTA g^−1^ to 5.85 mg EDTA g^−1^ in samples extracted by water and ethanol, respectively. 

Figure 11 presents the results of the Principal Component Analysis. The eigenvalues of axis 1 (6.90) and axis 2 (1.59) indicate the presence of two main gradients, within which the samples (water and ethanol extracts) are differentiated in terms of the chemical composition and antioxidant characteristics (Table 3). Axes 1 and 2 explain 65.35% of the variability (55.09%—axis 1, 12.26%—axis 2). These data prove that the analyzed secondary metabolites and antioxidant properties correlated with these axes are greatly important for interpreting the differentiation and correlations between these parameters. The main variation gradient extends along axis 1. The concentrations of total phenolics, total flavonoids, hyperoside, and methylarbutin are clearly positively correlated with axis 1, whereas corilagin, hydroquinone, and DPPH are correlated negatively. The concentrations of picein and arbutin are positively correlated with axis 2. In the ordination space, two groups of samples can be distinguished (Figure 11, Table 3). The WE are located on the left site. The correlation between secondary metabolites in the WE and antioxidant activity parameters shows a positive impact of the total phenolic concentration on ABTS and RP and a positive impact of the hydroquinone on ABTS, RP, and CHEL (Figure 11, Table 4). In turn, the EE create the second group and are located in the right part of the ordination space (Figure 11). The correlation between the studied metabolites and antioxidant activity parameters within this group show a positive impact of the total phenolic concentration on ABTS, DPPH, and RP. In the case of the EE, no statistically significant impact of HQ on the antioxidant activity parameters was found (Figure 11, Table 5).

Some studies demonstrated that the localization of populations and different habitat conditions were the main source of variance in the concentrations of secondary metabolites and in vitro antioxidant activity [9,10]. The antioxidant potential of BL has been studied with the use of numerous chemical assays showing their very high antioxidant activity [28,29,31]. The present study confirmed this finding. Namely, irrespective of the type of solvent, the variability of the populations determined the chemical composition, and the total phenolic concentration was responsible for the antioxidant activity of the raw material. In turn, flavonoids are plant polyphenols found in vegetables, fruits, and plant-based beverages and are well known for their physiological antipyretic, analgesic, and anti-inflammatory activities [64]. Some studies have demonstrated that flavonoids have obvious anti-inflammation and anti-oxidative stress activities, which are highly beneficial in treating diabetic retinopathy and exert beneficial impacts in treatment of diabetic complications [65]. Although flavonoids are regarded as powerful antioxidants [5,66], no relationship between flavonoids and antioxidant parameters was observed in the presented study. This lack of contribution of flavonoids to antioxidant properties can be explained by the generally low share of total flavonoids, (WE 2.36–3.09 mg QE g^−1^; EE 3.21–3.88 mg QE g^−1^) in total phenolic compounds (WE 165.63–214.84 mg GAE g^−1^; EE 258.03–298.52 mg GAE g^−1^), which is insufficient to exert a significant effect on the antioxidant parameters. 

Researchers suggest that the solvent polarity and plant species variety affect the extractability of polyphenols and flavonoids [66,67]. In the present study, this thesis is confirmed, e.g., by the higher concentrations of hydroquinone and corilagin and the lower concentrations of other metabolites, excluding arbutin, in the water extracts in relation to the ethanol extracts.

## 3. Materials and Methods

### 3.1. Chemicals

Chemicals for total phenolic and antioxidant assays: Folin-Ciocalteu′s phenol reagent, aluminum chloride (AlCl_3_; ≥99%), ABTS (2,2’-azino-bis (3-ethylbenzothiazoline-6-sulphonic acid), DPPH (2,2-diphenyl-1-picrylhydrazyl), TPTZ (2,4,6-Tri(2-pyridyl)-s-triazine),ferrozine (3-(2-pyridyl)-5,6-diphenyl-1,2,4-triazine-4′,4′′-disulfonic acid sodium salt), EDTA (ethylenediaminetetraacetic acid disodium salt; ≥99%); Trolox (6-hydroxy-2,5,7,8-tetramethylchroman-2-carboxylic acid; ≥97%), quercetin (Q; 3,3′,4′,5,6-pentahydroxyflavone; ≥97%), gallic acid (GA; 3,4,5-trihydroxybenzoic acid; ≥98%); chromatographic standards: arbutin (4-Hydroxyphenyl-β-D-glucopyranoside; ≥98%), hydroquinone (1,4-Dihydroxybenzene; ≥99%), methylarbutin (4-methoxyphenyl β-D-glucopyranoside; ≥97%), picein (4-acetylphenyl β-D-glucopyranoside; ≥98%), corilagin (1-O-galloyl-3,6-hexahydroxydiphenol-β-D-glucopyranose; ≥96%),penta-O-galloyl-β-D-glucose; ≥96%), and hyperoside (3,3′,4′,5,7-pentahydroxyflavone 3-D-galactoside; ≥95.0%), as well as mobile phase components: acetonitrile and formic acid (HPLC grades) were purchased from Merck company (Merck KGaA, Darmstadt, Germany). All other chemicals were of analytical grade.

### 3.2. Habitat Characteristics and Plant Material

The field study was carried out in twelve bearberry populations from forests of mid-eastern Poland in the first ten days of September 2020. The bearberry forms dense patches and *Pinus sylvetris* was the dominant tree species in the forest communities. The coverage of bearberry plants was in the range of 80–90%, and accompanying species (*Melampyrum pratense*, *Festuca ovina*, *Dicranum scoparium* and *Pleurozium schreberi*)–accounted for 10–30%. The soils in the analyzed habitats are mainly podzols. These very acidic soils with a predominant sand fraction are characterized by a very low content of available forms of phosphorus, potassium and magnesium. In each of the 12 sites, three bearberry leaf samples (40 g) were collected within 25-m^2^ dense phytocoenoses. After convection drying in a laboratory drying/heating oven (Binder FD 53, Binder GmbH, Tuttlingen, Germany) at 40 °C for 48 h, the plant material was powdered in a laboratory grinder and sieved to pass through a 1 mm screen.

### 3.3. Extraction Procedure

#### 3.3.1. Ethanol Extracts

Powdered BL (0.5 g) were extracted with 50 mL of 70% (*v/v*) hydroethanolic solution. Ultrasonic assisted extraction was performed for 1 h in an ultrasonic water bath (model IS-5.5, Intersonic, Olsztyn, Poland) set at 40 °C (ultrasound frequency 35 kHz, power 100 W). The samples were centrifuged at 4500× *g* for 15 min at room temperature and the supernatants were filtered through filter paper discs. Before analyses, the extracts were stored in a laboratory freezer at –50 °C.

#### 3.3.2. Water Extracts (Infusions) 

The plant material (0.5 g) was poured with 50 mL of boiled distilled water and shaken for 15 min using a rotator. Then, the infusions were transferred to an ultrasonic water bath set at 40 °C and extraction was continued for 45 min. The samples were centrifuged at 4500× *g* for 15 min at room temperature and the supernatants were filtered through filter paper discs. Before analyses, the extracts were stored in a laboratory freezer at –50 °C.

### 3.4. Phytochemical Characterization

#### 3.4.1. Total Phenolic Concentration

The spectrophotometric assay for determination of the total phenolic concentration was performed using Folin–Ciocalteu reagent according to the method proposed by Singleton and Rossi [68], following the previously described procedure [10]. The absorbance of the reaction mixtures (765 nm) were measured using an Epoch 2 microplate reader (BioTek Instruments, Inc., Winooski, VT, USA) and the results were expressed as Gallic acid equivalents (GAE) in mg∙g^−1^ dry matter of plant material.

#### 3.4.2. Total Flavonoid Concentration

Total flavonoid concentration was determined spectrophotometrically according to the Lamaison and Carnart [69] method based on the reaction of flavonoids with Al^3+^ ions from aluminum chloride, following the previously described procedure [10]. The absorbances of the reaction mixtures (430 nm) were measured using an Epoch 2 microplate reader (BioTek Instruments, Inc., Winooski, VT, USA) and the results were expressed as quercetin equivalents (QE) in mg g^−1^ dry matter of plant material.

#### 3.4.3. High-Performance Liquid Chromatography (HPLC)

Chromatographic separation was carried out as described previously [10] using a Varian ProStar HPLC separation system (Varian Inc., Walnut Creek, CA, USA) equipped with a Gemini C18 column (250 mm × 4.6 mm, 110 Å, 5 μm) (Phenomenex, Torrance, CA, USA). Briefly, extracts were diluted 10-times with the corresponding solvent and filtered through a 0.2 µm syringe filter. An aliquot (20 µL) was injected into the column thermostatted at 25 °C. Formic acid (0.1%; *v*/*v*)-acidified ultrapure water (A) and acetonitrile (B) were applied as the mobile phase using the following gradient: 4% B (prerun), 4–22% B (0 min–25 min); 22–25% B (25 min–40 min); 25–100% B (40 min–50 min); 100% B (50 min–55 min); 100–4% B (55 min–60 min); 4% B (60 min–65 min) at a flow rate of 1 mL min^−1^. The quantification of arbutin, methylarbutin, picein, corilagin, and penta-O-galloyl-β-d-glucose 280 nm was performed at 280 nm, whereas hyperoside was determined at 350 nm. The results were expressed in mg g^−1^ dry matter of plant material.

#### 3.4.4. Antioxidant Activity

Antioxidant activity was evaluated based on the ABTS^•+^ scavenging activity (ABTS) [70], and DPPH^•^ scavenging activity (DPPH) [71], ferric reducing power (RP) [72], and ferrous chelating ability (CHEL) [73] following previously described procedures [10]. The absorbance of the reaction mixtures were measured using a microplate photometer (Epoch 2, BioTek Instruments, Inc., Winooski, VT, USA). The results were expressed as Trolox equivalents (TE) for ABTS, DPPH and FRAP or EDTA equivalents for CHEL in mg g^−1^ dry matter of plant material.

### 3.5. Statistical Analysis

After testing the data for normality and homoscedasticity, two-way analysis of variance ANOVA was performed and followed by subsequent Tukey’s test. The results were presented as average values and standard deviation, and the differences were considered significant at *p* < 0.05. Correlations (Pearson coefficient) were estimated as well. The statistical analyses were carried out using the Statistica 6.0 software (Stat. Soft, Inc., Krakow, Poland). Principal component analysis was applied to explain the relationships between the parameters and to show variability factors. Prior to the PCA, the data were centered and log-transformed. The analyses were carried out using the MVSP program version 3.1 [74].

## 4. Conclusions

The studied populations are a valuable source of phenolic compounds, especially arbutin. The analyzed BL meet the European Pharmacopoeia requirements regarding the concentration of arbutin (>70 mg g^−1^); hence, they can be classified as herbal material and can be used in pharmacy. Therefore, the investigated populations can be a potentially valuable source of plant material for cultivation and can be used in in vitro cultures and in biotechnological processes. The present research has shown that the simultaneous use of a greater number of solvents provides better characterization of the chemical profile of BL and significantly expands the knowledge of the availability and antioxidant activity of chemicals. The type of solvent used in the research exerted an effect on the chemical characteristics of the raw material and antioxidant properties of the extracts, with emphasis on the role of hydrophilic components. The WE from BL were characterized by significantly higher concentrations of hydroquinone and corilagin and significantly lower concentrations of other metabolites, excluding arbutin, in comparison with the EE. In the case of the WE, a significant effect of not only total phenolic compounds, but also hydroquinone, on the antioxidant parameters was observed, which indicates the solvent-related activity of these metabolites. A very wide chemical diversity and variability of *Uvae ursi folium* is observed in commercial products available on the market. There is also a large variety of bearberry chemical profiles in natural populations determined by environmental conditions. Therefore, it is suggested that special attention should be paid to the concentration of not only arbutin, but also hydroquinone. The latter metabolite, serving a very important function as an active bearberry ingredient, should be controlled not only in alcoholic extract, but also in WE, since BL are applied as infusions and decoctions.

## Figures and Tables

**Figure 1 molecules-27-02247-f001:**
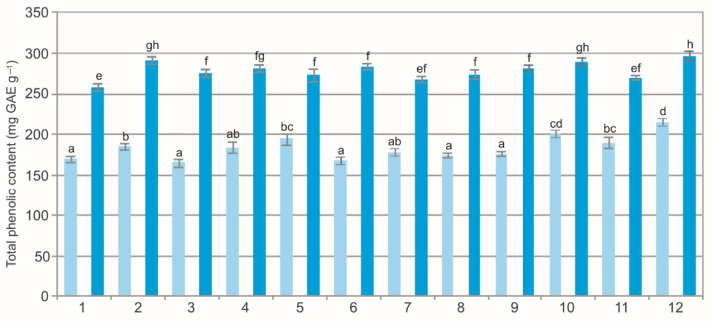
Total phenolic concentration in water (light blue) and ethanol (dark blue) leaf extracts from bearberry plants in different populations. The values designated by the different letters are significantly different (*p* < 0.05).

**Figure 2 molecules-27-02247-f002:**
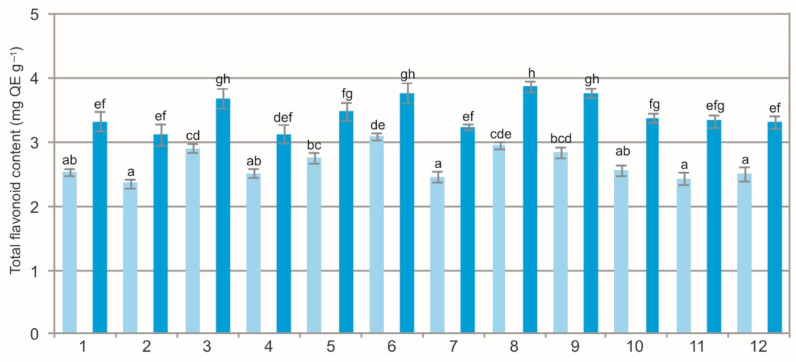
Total flavonoid concentration in water (light blue) and ethanol (dark blue) leaf extracts from bearberry plants in different populations. The values designated by the different letters are significantly different (*p* < 0.05).

**Figure 3 molecules-27-02247-f003:**
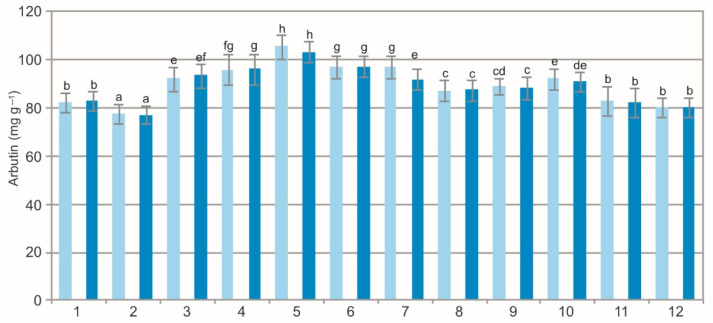
Arbutin concentration in water (light blue) and ethanol (dark blue) leaf extracts from bearberry plants in different populations. The values designated by the different letters are significantly different (*p* < 0.05).

**Figure 4 molecules-27-02247-f004:**
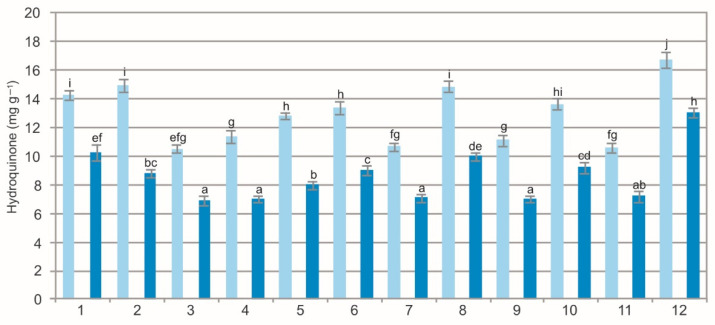
Hydroquinone concentration in water (light blue) and ethanol (dark blue) leaf extracts from bearberry plants in different populations. The values designated by the different letters are significantly different (*p* < 0.05).

**Figure 5 molecules-27-02247-f005:**
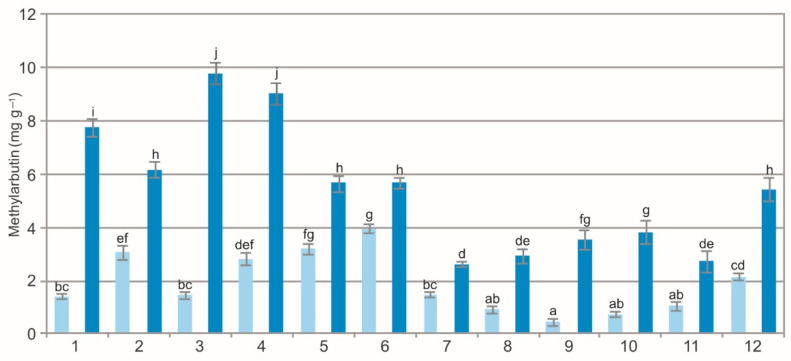
Methylarbutin concentration in water (light blue) and ethanol (dark blue) leaf extracts from bearberry plants in different populations. The values designated by the different letters are significantly different (*p* < 0.05).

**Figure 6 molecules-27-02247-f006:**
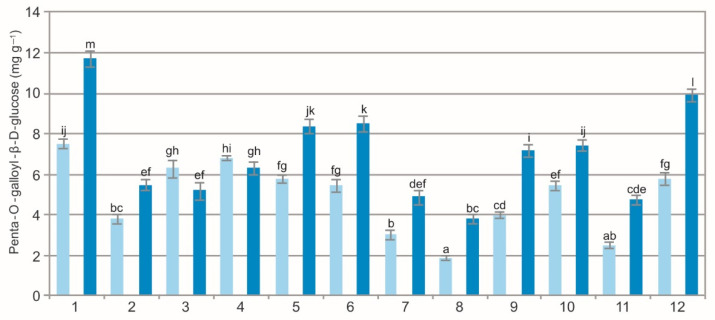
Penta-O-galloyl-β-d-glucose concentration in water (light blue) and ethanol (dark blue) leaf extracts from bearberry plants in different populations. The values designated by the different letters are significantly different (*p* < 0.05).

**Figure 7 molecules-27-02247-f007:**
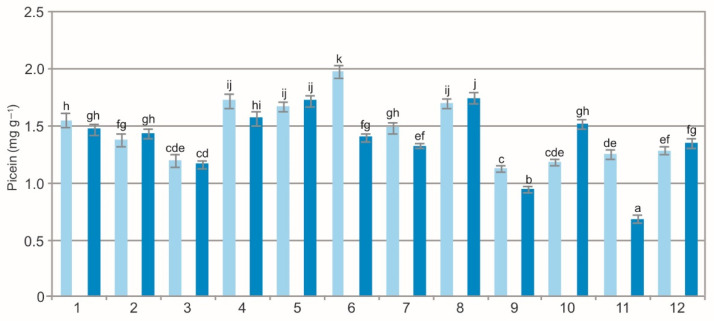
Picein concentration in water (light blue) and ethanol (dark blue) leaf extracts from bearberry plants in different populations. The values designated by the different letters are significantly different (*p* < 0.05).

**Figure 8 molecules-27-02247-f008:**
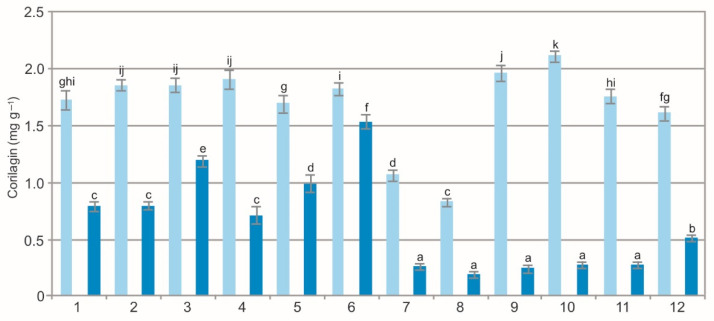
Corilagin concentration in water (light blue) and ethanol (dark blue) leaf extracts from bearberry plants in different populations. The values designated by the different letters are significantly different (*p* < 0.05).

**Figure 9 molecules-27-02247-f009:**
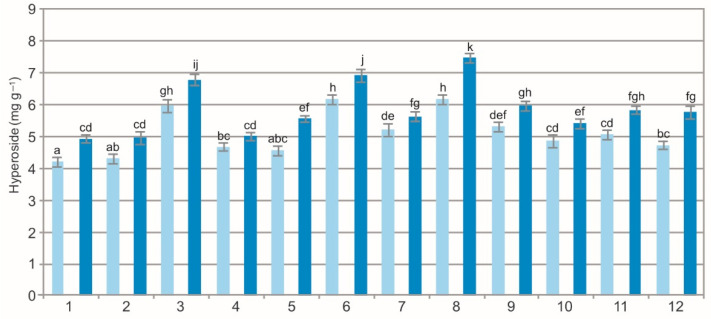
Hyperoside concentration in water (light blue) and ethanol (dark blue) leaf extracts from bearberry plants in different populations. The values designated by the different letters are significantly different (*p* < 0.05).

**Figure 10 molecules-27-02247-f010:**
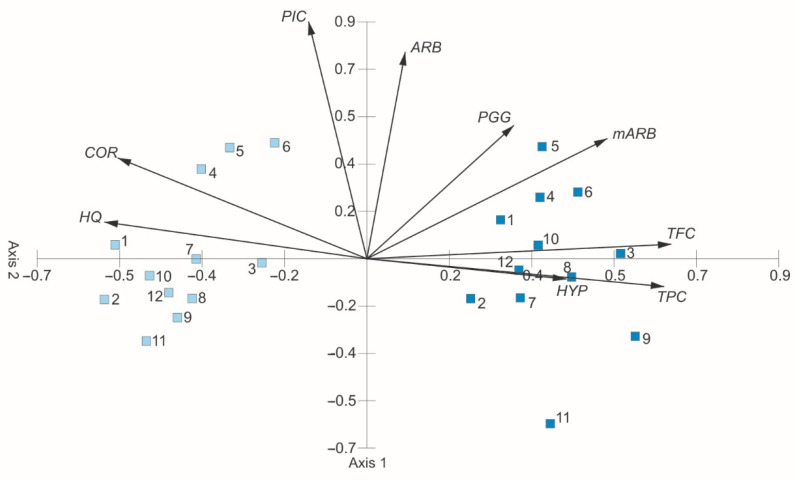
PCA ordination on the basis of the chemical composition of bearberry leaf extracts. TPC—total phenolic concentration, TFC—total flavonoid concentration, ARB—arbutin, HQ—hydroquinone, mARB—methylarbutin, PGG—penta-O-galloyl-β-d-glucose, PIC—picein, COR—corilagin, HYP—hyperoside.

**Figure 11 molecules-27-02247-f011:**
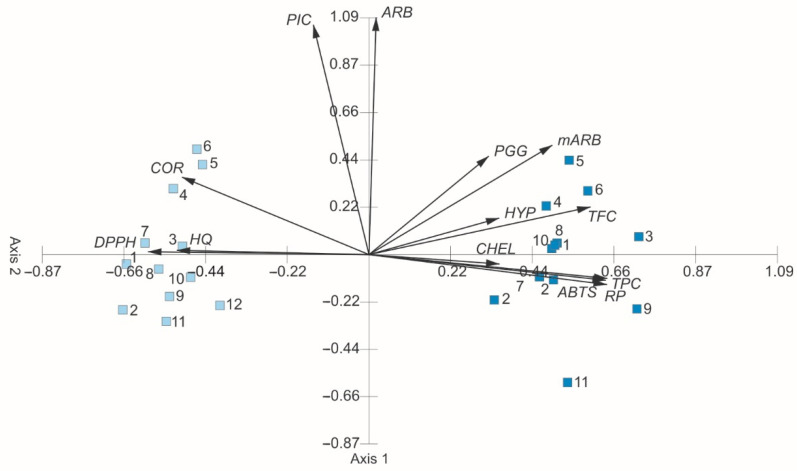
PCA ordination on the basis of the chemical composition of bearberry leaf extracts and antioxidant activity parameters. TPC—total phenolic concentration, TFC—total flavonoid concentration, ARB—arbutin, HQ—hydroquinone, mARB—methylarbutin, PGG—penta-O-galloyl-β-d-glucose, PIC—picein, COR—corilagin, HYP—hyperoside, ABTS—ABTS^•+^ scavenging activity, DPPH—DPPH^•^ scavenging activity, RP—reducing power, CHEL—chelating ability.

**Table 1 molecules-27-02247-t001:** Results of PCA based on the secondary metabolite composition of water and ethanol bearberry leaf extracts. TPC—total phenolic concentration, TFC—total flavonoid concentration, ARB—arbutin, HQ—hydroquinone, mARB—methylarbutin, PGG—penta-O-galloyl-β-d-glucose, PIC—picein, COR—corilagin, HYP—hyperoside.

	Axis 1	Axis 2
Eigenvalues	4.08	1.58
Percentage	45.38	17.50
TPC	0.450	−0.073
TFC	0.461	0.038
ARB	0.058	0.544
HQ	−0.397	0.096
mARB	0.364	0.317
PGG	0.222	0.351
PIC	−0.088	0.625
COR	−0.377	0.266
HYP	0.302	−0.054

**Table 2 molecules-27-02247-t002:** Antioxidant activity parameters of water (WE) and ethanol (EE) leaf extracts. ABTS—ABTS^•+^ scavenging activity, DPPH—DPPH^•^ scavenging activity, RP—reducing power, CHEL—chelating ability. The values designated by the different letters are significantly different (*p* < 0.05).

	ABTS		DPPH		RP		CHEL	
No	Mean	SD	Mean	SD	Mean	SD	Mean	SD
1 ^WE^	201.16 ^ab^	15.96	683.86 ^h^	16.84	251.42 ^a^	3.48	4.49 ^cde^	0.15
1 ^EE^	511.65 ^f^	17.26	387.71 ^ab^	25.22	359.70 ^ef^	3.72	5.60 ^hi^	0.14
2 ^WE^	243.57 ^cd^	17.16	787.67 ^ij^	37.84	270.66 ^bc^	7.62	4.23 ^bc^	0.18
2 ^EE^	548.77 ^g^	16.86	423.62 ^ab^	10.25	401.23 ^i^	5.51	3.56 ^a^	0.16
3 ^WE^	173.48 ^a^	13.44	723.18 ^hi^	28.23	245.37 ^a^	5.47	4.17 ^bc^	0.13
3 ^EE^	503.52 ^f^	15.25	361.82 ^a^	2.57	374.29 ^fgh^	5.00	5.85 ^i^	0.15
4 ^WE^	211.48 ^bc^	19.10	701.66 ^h^	22.39	259.05 ^abc^	6.72	4.31 ^bc^	0.10
4 ^EE^	560.28 ^g^	18.58	401.97 ^ab^	12.74	375.60 ^gh^	8.82	4.19 ^bc^	0.13
5 ^WE^	226.39 ^bc^	13.83	519.62 ^de^	15.96	257.10 ^ab^	5.94	3.74 ^a^	0.09
5 ^EE^	566.53 ^gh^	14.88	381.75 ^a^	25.89	377.82 ^gh^	7.94	4.90 ^gh^	0.14
6 ^WE^	201.16 ^ab^	13.75	821.78 ^j^	20.06	257.23 ^ab^	4.51	4.46 ^cde^	0.11
6 ^EE^	543.71 ^g^	19.05	372.18 ^ab^	13.52	376.51 ^gh^	7.64	4.78 ^ef^	0.16
7 ^WE^	194.45 ^ab^	13.95	665.70 ^gh^	27.64	247.20 ^a^	4.72	3.83 ^ab^	0.11
7 ^EE^	547.18 ^g^	13.94	377.33 ^ab^	11.12	353.26 ^e^	3.05	4.60 ^def^	0.09
8 ^WE^	191.80 _ab_	15.87	677.25 ^h^	15.40	246.86 ^a^	6.01	4.53 ^cdef^	0.15
8 ^EE^	537.54 ^g^	11.98	385.79 ^ab^	13.48	361.20 ^efg^	5.93	4.17 ^bc^	0.14
9 ^WE^	195.43 ^ab^	17.65	605.27 ^fg^	28.95	261.21 ^abc^	2.07	4.33 ^cd^	0.16
9 ^EE^	561.95 ^g^	17.90	412.02 ^ab^	13.52	377.75 ^gh^	7.83	5.34 ^gh^	0.15
10 ^WE^	260.24 ^d^	13.96	572.98 ^ef^	29.56	275.74 ^b^	5.04	4.53 ^cdef^	0.08
10 ^EE^	596.31 ^h^	18.66	447.16 ^bc^	21.97	383.76 ^h^	5.58	4.61 ^cdef^	0.14
11 ^WE^	245.62 ^cd^	19.09	490.59 ^cd^	16.02	259.95 ^abc^	5.76	3.71 ^a^	0.09
11 ^EE^	556.75 ^g^	10.32	415.57 ^ab^	27.18	365.13 ^efg^	6.00	4.82 ^ef^	0.15
12 ^WE^	319.22 ^e^	15.25	723.90 ^hi^	13.59	296.20 ^d^	3.89	4.96 ^fg^	0.11
12 ^EE^	643.71 ^i^	19.91	516.28 ^de^	13.32	404.89 ^i^	5.77	4.89 ^ef^	0.14

**Table 3 molecules-27-02247-t003:** Results of PCA based on the secondary metabolite composition of bearberry water and ethanol leaf extracts and antioxidant properties. TPC—total phenolic concentration, TFC—total flavonoid concentration, ARB—arbutin, HQ—hydroquinone, mARB—methylarbutin, PGG—penta-O-galloyl-β-d-glucose, PIC—picein, COR—corilagin, HYP—hyperoside, ABTS—ABTS^•+^ scavenging activity, DPPH—DPPH^•^ scavenging activity, RP—reducing power, CHEL—chelating ability.

	Axis 1	Axis 2
Eigenvalues	6.90	1.59
Percentage	53.09	12.26
TPC	0.364	−0.068
TFC	0.340	0.126
ARB	0.011	0.628
HQ	−0.295	0.011
mARB	0.282	0.290
PGG	0.184	0.261
PIC	−0.085	0.609
COR	−0.287	0.205
HYP	0.200	0.097
ABTS	0.365	−0.080
DPPH	−0.339	0.008
RP	0.366	−0.063
CHEL	0.200	−0.025

**Table 4 molecules-27-02247-t004:** Relationships (Pearson correlation coefficients) between the secondary metabolites of bearberry WE and antioxidant parameters. TPC—total phenolic concentration, TFC—total flavonoid concentration, ARB—arbutin, HQ—hydroquinone, mARB—methylarbutin, PGG—penta-O-galloyl-β-d-glucose, PIC—picein, COR—corilagin, HYP—hyperoside, ABTS—ABTS^•+^ scavenging activity, DPPH—DPPH^•^ scavenging activity, RP—reducing power, CHEL—chelating ability. * *p* < 0.05; ** *p* < 0.01; *** *p* < 0.001.

	ABTS	DPPH	RP	CHEL
TPC	0.937 ***	−0.362	0.855 ***	0.191
TFC	−0.523	0.199	−0.402	0.156
ARB	−0.401	−0.218	−0.407	−0.415
HQ	0.580 *	0.378	0.592 *	0.721 **
mARB	0.087	0.496	0.100	−0.047
PGG	0.055	0.238	0.153	0.316
PIC	−0.298	0.370	−0.345	0.002
COR	0.218	−0.061	0.369	0.064
HYP	−0.492	0.233	−0.430	0.067

**Table 5 molecules-27-02247-t005:** Relationships (Pearson correlation coefficients) between the secondary metabolites of bearberry EE and antioxidant parameters. TPC—total phenolic concentration, TFC—total flavonoid concentration, ARB—arbutin, HQ—hydroquinone, mARB—methylarbutin, PGG—penta-O-galloyl-β-d-glucose, PIC—picein, COR—corilagin, HYP—hyperoside, ABTS—ABTS^•+^ scavenging activity, DPPH—DPPH^•^ scavenging activity, RP—reducing power, CHEL—chelating ability. ** *p* < 0.01; *** *p* < 0.001.

	ABTS	DPPH	RP	CHEL
TPC	0.781 **	0.786 **	0.896 ***	−0.338
TFC	−0.262	−0.353	−0.213	0.378
ARB	−0.152	−0.539	−0.284	0.195
HQ	0.352	0.501	0.308	−0.114
mARB	−0.335	−0.194	0.221	0.276
PGG	0.250	0.301	0.239	0.423
PIC	0.041	−0.089	0.073	−0.376
COR	−0.347	−0.387	0.199	0.222
HYP	−0.230	−0.316	−0.235	0.171

## Data Availability

Not applicable.

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
