# Peer review of "Variation in Population and Solvents as Factors Determining the Chemical Composition and Antioxidant Potential of *Arctostaphylos uva-ursi* (L.) Spreng. Leaf Extracts"

_molecules, 2022, doi:10.3390/molecules27072247_

Round 1

Reviewer 1 Report

molecules-1639753

The manuscript Molecules 1639753, submitted by Sugier et al., presents the results of a research that aimed to characterize the variability of the phytochemical composition and antioxidant activity polar bearberry extracts. The plant material collected was from different natural populations. The methodology consisted on the determination of total phenolic and flavonoid concentration; chromatographic profiling by HPLC, and ABTS/DPPH antioxidant activity. A total of 12 samples were evaluated.

The manuscript is very prolix, with many literature citations that do not help in the manuscript flow. The manuscript deserves major revision, and I will point some of the points that I suggest to be addressed by the authors.

Abstract

Lines 17-19: I suggest to move the following part to the end of the abstract: “The results presented in this paper can contribute to 17 appropriate selection of plant material for pharmaceutical, cosmetic, and food industries, with 18 special emphasis on the antioxidant activity of different types of extracts”

In the same way, this part “The studied bearberry leaves can be classified as a suitable herbal material for use in phar- 22 macy; therefore, the investigated populations can be a potentially valuable source of plant material 23 for cultivation and can be used in in vitro cultures and in biotechnological processes.” (lines 22 to 24) should be placed before presenting the objectives (line 19).

Line 21. Please, cite the number of samples used in the work.

In general, it should be useful to add some quantitative data, because all the results are presented qualitatively “higher” “lower concentration”, “significant effect”, etc. Please, be more specific.

Introduction.

Lines 37 to 39. The authors should give recent references when they say “has 37 been growing rapidly in the world”, because two out the three references cited were from 2003 and 2004. The same for the following references that would be updated with more recent examples.

Line 61. What does “(B)” mean? If it is an abbreviation, should be BL, as used in line 65. In the manuscript, “B” was used to indicate the mobile phase for chromatographic elution.

Line 80: Please, consider replacing “concentration” by “presence”,

Line 81. rare instead of rarely

Lines 89 and followings. This part is very wordy. Please, rephrase.

Line 94: I suggest: “the use of water as solvent” instead of “water used as a solvent”

Line 97: I suggest: “the use of different solvents” instead of “, the use of more types of solvents”

Results

Line 76. Please, correct because you have Results and Discussion together

The results were duly evaluated in terms of statistical relevance.

In general, I found the results and discussion section very prolix and confusing. For example, the authors say (lines 149-152) that “The concentration of arbutin in plants varies depending on the species, parts of the plant, development stages, and harvest season [11,41]. The arbutin production efficiency also varies depending on the extraction method [12,42,43]”. These information are well-known; it is not a characteristic unique of arbutin. Therefore, it is not necessary to give 5 references to prove something that natural products scientists know. [this was the prolix part]

Then, the authors continue: “This fact was not confirmed in the present study.” Therefore, what is the point of proving with the literature that the concentration varies, if you have conflicting results? [this is the part confusing]

Of course, results are results, but you could say something like that: Although the concentration of phytoconstituents usually vary according to seasonal and other specific characteristics (part of the plant, species, etc), and the yield depends on the extraction method, this variation was not observed in the present study. Then, please, explain why. The extracts you worked with are both polar, therefore, the small difference of polarity did not influence the extract composition. It should be different comparing, for example, ethyl acetate and aqueous extracts. Since the sample collection was in the same period (line 411) and in the same region (line 411), this can justify the fact of having similar results.

 Lines 142, 164, 185, etc. Please, add the information here again. All graphs, tables and figures must be self-explanatory.

Lines 200 and 201. “mg” is repeated a few times. Please, correct.

Lines 381-382. The authors explain, about flavonoids, that “no relationship between this group of secondary metabolites and antioxidant parameters was observed”, supposably because of the polar solvents used for extraction. This should be better discussed, because at least the ethanol extract has high concentration of flavonoids and flavonoids are indubitably antioxidants. As I pointed before, there are unnecessary information here, such as those from lines 376 to 381. Writing too much about peripheric subjects so not help understanding your results.

I would advise to clear other peripheric information from the manuscript.

Materials and Methods

Line 424. Please, add a space between the value (50) and the unity (°C) here to meet the standard international way to present the temperature; Do the same in other places where the space is missing.

Author Response

Responses to the remarks of Reviewer 1

We would like to thank the reviewer for the valuable comments that helped us to significantly improve the manuscript. The detailed responses to the reviewer’s comments are given below.

The manuscript Molecules 1639753, submitted by Sugier et al., presents the results of a research that aimed to characterize the variability of the phytochemical composition and antioxidant activity polar bearberry extracts. The plant material collected was from different natural populations. The methodology consisted on the determination of total phenolic and flavonoid concentration; chromatographic profiling by HPLC, and ABTS/DPPH antioxidant activity. A total of 12 samples were evaluated.

The manuscript is very prolix, with many literature citations that do not help in the manuscript flow. The manuscript deserves major revision, and I will point some of the points that I suggest to be addressed by the authors.

Abstract

Lines 17-19: I suggest to move the following part to the end of the abstract: “The results presented in this paper can contribute to appropriate selection of plant material for pharmaceutical, cosmetic, and food industries, with special emphasis on the antioxidant activity of different types of extracts”

- Thank you very much for this comment. Corrected in accordance with the recommendations of the reviewer.

In the same way, this part “The studied bearberry leaves can be classified as a suitable herbal material for use in pharmacy; therefore, the investigated populations can be a potentially valuable source of plant material for cultivation and can be used in in vitro cultures and in biotechnological processes.” (lines 22 to 24) should be placed before presenting the objectives (line 19).

- Thank you very much for this comment. Corrected in accordance with the recommendations of the reviewer.

Line 21. Please, cite the number of samples used in the work.

- The number of samples used in the work was cited.

In general, it should be useful to add some quantitative data, because all the results are presented qualitatively “higher” “lower concentration”, “significant effect”, etc. Please, be more specific.

- Corrected in accordance with the recommendations of the reviewer. Some quantitative data were added.

Introduction.

Lines 37 to 39. The authors should give recent references when they say “has been growing rapidly in the world”, because two out the three references cited were from 2003 and 2004. The same for the following references that would be updated with more recent examples.

- Thank you very much for this comment. New articles have been added and cited. Cited literature was renumbered.

Line 61. What does “(B)” mean? If it is an abbreviation, should be BL, as used in line 65. In the manuscript, “B” was used to indicate the mobile phase for chromatographic elution.

- Corrected in accordance with the recommendations of the reviewer.

Line 80: Please, consider replacing “concentration” by “presence”,

- Corrected in accordance with the recommendations of the reviewer.

Line 81. rare instead of rarely

- Corrected in accordance with the recommendations of the reviewer.

Lines 89 and followings. This part is very wordy. Please, rephrase.

- Corrected in accordance with the recommendations of the reviewer.

Line 94: I suggest: “the use of water as solvent” instead of “water used as a solvent”

- Corrected in accordance with the recommendations of the reviewer.

Line 97: I suggest: “the use of different solvents” instead of “, the use of more types of solvents”

- Corrected in accordance with the recommendations of the reviewer.

Results

Line 76. Please, correct because you have Results and Discussion together

- Thank you very much for this comment. Corrected in accordance with the recommendations of the reviewer.

The results were duly evaluated in terms of statistical relevance.

In general, I found the results and discussion section very prolix and confusing. For example, the authors say (lines 149-152) that “The concentration of arbutin in plants varies depending on the species, parts of the plant, development stages, and harvest season [11,41]. The arbutin production efficiency also varies depending on the extraction method [12,42,43]”. These information are well-known; it is not a characteristic unique of arbutin. Therefore, it is not necessary to give 5 references to prove something that natural products scientists know. [this was the prolix part]

- Thank you very much for this comment. Corrected in accordance with the recommendations of the reviewer.

Then, the authors continue: “This fact was not confirmed in the present study.” Therefore, what is the point of proving with the literature that the concentration varies, if you have conflicting results? [this is the part confusing]

- Thank you very much for this comment. Corrected in accordance with the recommendations of the reviewer.

Of course, results are results, but you could say something like that: Although the concentration of phytoconstituents usually vary according to seasonal and other specific characteristics (part of the plant, species, etc), and the yield depends on the extraction method, this variation was not observed in the present study. Then, please, explain why. The extracts you worked with are both polar, therefore, the small difference of polarity did not influence the extract composition. It should be different comparing, for example, ethyl acetate and aqueous extracts. Since the sample collection was in the same period (line 411) and in the same region (line 411), this can justify the fact of having similar results.

- Thank you very much for this comment. Corrected in accordance with the recommendations of the reviewer.

 Lines 142, 164, 185, etc. Please, add the information here again. All graphs, tables and figures must be self-explanatory.

Information has been added to all tables and figures.

Lines 200 and 201. “mg” is repeated a few times. Please, correct.

- Corrected in accordance with the recommendations of the reviewer.

Lines 381-382. The authors explain, about flavonoids, that “no relationship between this group of secondary metabolites and antioxidant parameters was observed”, supposably because of the polar solvents used for extraction. This should be better discussed, because at least the ethanol extract has high concentration of flavonoids and flavonoids are indubitably antioxidants. As I pointed before, there are unnecessary information here, such as those from lines 376 to 381. Writing too much about peripheric subjects so not help understanding your results.

I would advise to clear other peripheric information from the manuscript.

- Corrected in accordance with the recommendations of the reviewer.

Materials and Methods

Line 424. Please, add a space between the value (50) and the unity (°C) here to meet the standard international way to present the temperature; Do the same in other places where the space is missing.

- Thank you very much for this comment. Corrected in accordance with the recommendations of the reviewer.

Reviewer 2 Report

  1. Abstract: in line 27 the abbreviation WE appears, without indicating what it refers to, since it is the first time it appears in the text, it should be indicated in parentheses.
  2. Line 61: bearberry leaves (B), the abbreviation is BL in the text.
  3. Line 87: (BL): the parenthesis must be eliminated.
  4. A better description of the 12 populations studied is needed.

Author Response

Responses to the remarks of Reviewer 2

We would like to thank the reviewer for the valuable comments that helped us to significantly improve the manuscript.

Abstract: in line 27 the abbreviation WE appears, without indicating what it refers to, since it is the first time it appears in the text, it should be indicated in parentheses.

- Thank you very much for this comment. Corrected in accordance with the recommendations of the reviewer.

Line 61: bearberry leaves (B), the abbreviation is BL in the text.

- Corrected in accordance with the recommendations of the reviewer.

Line 87: (BL): the parenthesis must be eliminated.

- Corrected in accordance with the recommendations of the reviewer.

A better description of the 12 populations studied is needed.

- Thank you very much for this comment. Corrected in accordance with the recommendations of the reviewer.

Reviewer 3 Report

The authors did a good work from an experimental point of view, and I recommend the article for publication after some minor revisions.

More specific:

L13: The abstract is very extensive.

L27: Explain the abbreviation WE.

L61: What is (B)? You mean (BL)?

L68: Explain the abbreviation PGG.

L87: Without parentheses.

L390: Justify the text in sub-section paragraph.

L415: After drying… How?

L421: Provide more details on ultrasonic extraction. Machine, model and company are missing.

L429: Why was the extraction procedure not the same as above?

L461: PGG 280 nm…?

General comment: I’m very confused and worried that arbutin may be toxic in high doses and more that is metabolized to form hydroquinone, a potential liver toxin.

Author Response

Responses to the remarks of Reviewer 3

We would like to thank the reviewer for the valuable comments that helped us to significantly improve the manuscript.

The authors did a good work from an experimental point of view, and I recommend the article for publication after some minor revisions.

More specific:

L13: The abstract is very extensive.

- Thank you very much for this comment. Corrected in accordance with the recommendations of the reviewer.

L27: Explain the abbreviation WE.

- Corrected in accordance with the recommendations of the reviewer. The abbreviation was explained.

L61: What is (B)? You mean (BL)?

- Corrected in accordance with the recommendations of the reviewer.

L68: Explain the abbreviation PGG.

- Corrected in accordance with the recommendations of the reviewer.

L87: Without parentheses.

- Corrected in accordance with the recommendations of the reviewer. Parentheses have been removed.

L390: Justify the text in sub-section paragraph.

- Corrected in accordance with the recommendations of the reviewer. The text was formatted.

L415: After drying… How?

- Corrected in accordance with the recommendations of the reviewer. This part has been completed.

L421: Provide more details on ultrasonic extraction. Machine, model and company are missing.

- Corrected in accordance with the recommendations of the reviewer. This part has been completed.

L429: Why was the extraction procedure not the same as above?

Since water infusions are one of the most commonly used medicinal and domestic preparations of bearberry leaves, we decided to use brewing in the first step of extraction. After the hydrothermal treatment, to provide easier release of the compounds from the plant matrix, ultrasound treatment was applied as in the case of ethanolic extracts.

L461: PGG 280 nm…?

- Corrected in accordance with the recommendations of the reviewer.

General comment: I’m very confused and worried that arbutin may be toxic in high doses and more that is metabolized to form hydroquinone, a potential liver toxin.

Round 2

Reviewer 1 Report

The manuscript was improved and I recommend to accept in the current form.